# An Overview of Lutein in the Lipid Membrane

**DOI:** 10.3390/ijms241612948

**Published:** 2023-08-18

**Authors:** Justyna Widomska, Witold K. Subczynski, Renata Welc-Stanowska, Rafal Luchowski

**Affiliations:** 1Department of Biophysics, Medical University of Lublin, 20-090 Lublin, Poland; 2Department of Biophysics, Medical College on Wisconsin, Milwaukee, WI 53226, USA; subczyn@mcw.edu; 3Institute of Agrophysics, Polish Academy of Sciences, 20-290 Lublin, Poland; r.welc@ipan.lublin.pl; 4Department of Biophysics, Institute of Physics, Maria Curie-Sklodowska University, 20-031 Lublin, Poland; rafal.luchowski@umcs.pl

**Keywords:** lutein, macular pigments, lipid membrane

## Abstract

Lutein, zeaxanthin, and *meso*-zeaxanthin (a steroisomer of zeaxanthin) are macular pigments. They modify the physical properties of the lipid bilayers in a manner similar to cholesterol. It is not clear if these pigments are directly present in the lipid phase of the membranes, or if they form complexes with specific membrane proteins that retain them in high amounts in the correct place in the retina. The high content of macular pigments in the Henle fiber layer indicates that a portion of the lutein and zeaxanthin should not only be bound to the specific proteins but also directly dissolved in the lipid membranes. This high concentration in the prereceptoral region of the retina is effective for blue-light filtration. Understanding the basic mechanisms of these actions is necessary to better understand the carotenoid–membrane interaction and how carotenoids affect membrane physical properties—such as fluidity, polarity, and order—in relation to membrane structure and membrane dynamics. This review focuses on the properties of lutein.

## 1. Introduction

Lutein (Lut), zeaxanthin (Zea), and *meso*-zeaxanthin (*meso*-Zea) are major pigments of the yellow spot in the human retina [1,2,3]. The carotenoids Lut and Zea are obtained from food, whereas the third carotenoid, *meso*-Zea, is almost absent in the human diet and plasma [4]. *Meso*-Zea appears in the retina as a result of the conversion of Lut into *meso*-Zea directly in the retinal pigment epithelium (RPE) [5]. The spatial distribution of all macular pigments is well known and has been mapped using Raman imaging [6]. The two most important conclusions for these studies are that (1) the ratio of Lut/Zea changes eccentricity, and (2) the Lut/Zea+*meso*-Zea ratio is lowest in the foveola center [6]. The distribution of macular pigments in the retinal layers and their subcellular localizations is not fully understood. Most macular carotenoids are concentrated in the outer plexiform layer at the fovea and partly in the inner plexiform layer [7,8]. According to the Gass hypothesis, Müller cell cones may also be reservoirs for macular carotenoids [9]. Moreover, macular carotenoids are detected in small concentrations in the rod outer segment and RPE layer [10,11]. It is widely accepted that the ratio of Lut/Zea+*meso*-Zea is not the same for adult retinas and retinas from young donors, aged 0 to 3 years [2,12]. Bone et al. [2] pointed out that for young retinas that are not fully matured, Lut and Zea accumulate differently from in matured retinas with higher concentrations of Lut. Similar to the retina, in human brain tissue, xanthophylls are the predominant carotenoids. It has been shown that Lut constitutes ~60% [13] of total brain carotenoid content in infant neural tissue, whereas it accounts for a lower value of ~30% [14] in adult neural tissue. These results strongly suggest preferential uptake of carotenoids with two hydroxyl groups into brain tissue, similar to that in retinal tissue [15]. 

Our last paper focused on Zea [16] because it is selectively accumulated in the fovea. In this review, we consider the Lut–lipid interaction. Structurally, Lut is different from Zea. Both Zea and *meso*-Zea contain two β-ring end groups, whereas Lut contains one β-ring and one ε-ring (see Figure 1). Lut has 10 conjugated double bonds (N = 10), whereas Zea and *meso*-Zea have longer conjugated double bond systems (N = 11). Finally, the spatial orientation of the hydroxyl group on the C3′ chiral position of the *meso*-Zea molecule is the S spatial orientation (3R, 3′S configuration). In contrast with *meso*-Zea, Zea has an R spatial orientation (3R, 3′R configuration). Lut, on the other hand, has three chiral centers and a 3R, 3′R, 6′R configuration. That is to say that Zea and *meso*-Zea are structurally similar. Lut has a shorter polyene chain with one different ring compared to Zea and *meso*-Zea. Finally, almost nothing is known about the form in which they exist in retinal cells, whether they are bound to proteins or are a free component of the lipid matrix. Two proteins have been identified as macular pigment uptake proteins: a pi isoform of human glutathione S-transferase (GSTP1) [17] for Zea and StAR-related lipid-transfer protein 3 (StARD3) as a Lut-binding protein [18]. The question remains as to whether these proteins only transport macular carotenoids to their proper place or maybe store them in these specific membranous structures. The Henle fiber layer contains a large amount of macular pigment, which indicates that some part of it should be dissolved directly in the lipid matrix. This review summarizes Lut–lipid model studies and the unique Lut–membrane interaction that distinguishes Lut and Zea from other carotenoids available in the human diet. 

## 2. Horizontal or Vertical Orientation of Lut in the Lipid Bilayer?

Macular pigments are xanthophylls with two hydroxyl groups on the ends of the molecules (Figure 1); therefore, they are more hydrophilic than other plasma carotenoids like α-carotene, β-carotene, lycopene, and β-cryptoxanthin. These two hydroxyl groups can potentially be anchored in the opposite polar regions of the membrane, span the hydrophobic core of the lipid bilayer, and promote lateral packing. Incorporating the rigid, rod-like polar carotenoid molecules into the membrane enhances extended *trans*-conformation of the alkyl chains, decreases free space in the bilayer center, separates the phospholipid head groups, and decreases interactions between them. It is widely accepted that Zea adopts a vertical orientation in the lipid bilayer [16,19,20,21,22]. In contrast, the molecular orientation of Lut is not unambiguous. Two orientations of the Lut molecule—horizontal and vertical to the lipid bilayer—were previously reported by Sujak et al. [21]. They used linear dichroism analysis to calculate the mean orientation of the dipole transition moment of the xanthophyll molecule incorporated into the lipid multi-bilayers formed by egg lecithin (EYPC). A relatively large orientation angle of 67° [21] between the transition dipole and the axis normal to the plane of the lipid membrane was found in the case of Lut and was interpreted as a representation of the two pools of Lut vertically and parallelly oriented. The average orientation angle between the axis normal to the plane of the EYPC bilayer and the transition dipole of the Zea molecule obtained from the same studies was much smaller (33°) [21]. In summary, these lipid model studies demonstrated that Lut and Zea have different orientations in model lipid membranes. Krinsky was the first to question the parallel orientation of Lut. He speculated that, other than for Zea, this unusual result was caused by the formation of *cis*-isomers or dehydration of some of the Lut molecules [23]. Moreover, electron paramagnetic resonance (EPR) studies do not appear to support two orientations of Lut in the lipid bilayer. Both macular carotenoids reduce the fluidity of the membrane interior and organize the alkyl chains, which indicates organization across the lipid bilayer. There is also no difference in the effects on the order of the hydrophobic core of the membrane. Lut and Zea decrease the fluidity of this membrane region as well as increasing the order of alkyl chains in the membrane center [24]. Finally, hydrophobicity profiles across the lipid membrane for these two polar carotenoids show a typical bell-like shape with a gradual increase in hydrophobicity toward the center of the bilayer [25]. None of these effects on the physical properties of the lipid membranes indicate significant differences between these two carotenoids in the model lipid membrane formed from phosphatidylcholine [26]. Based on our own EPR measurements, we assume that Lut and Zea adopt an orientation perpendicular to the plane of the membrane. The preferred orientations of Lut in the phosphatidylcholine bilayers were studied by M. Pasenkiewicz-Gierula et al. using molecular modeling methodology [27]. The molecular dynamics results show that Lut intercalates into the lipid bilayer preferentially from the β-ring-side and adopts a mostly vertical position [28]. However, the thickness of phosphatidylcholine bilayers fluctuates and may be larger than the C3-C3′ distance of the Lut molecule, which may promote cross-membrane rotation and the horizontal position of Lut in the bilayer [22,27]. As mentioned, the effect of Lut on the physical properties of the membrane is strong because Lut is a dipolar carotenoid, but this effect depends on membrane thickness and is weak for thicker membranes (18-carbon phosphatidylcholine and 22-carbon phosphatidylcholine), which may also indicate the horizontal orientation of Lut in these bilayers [26].

In the model system comprising supported lipid bilayers, the orientation angle depends on the number of stacked bilayers. This is most likely due to the fact that a certain fraction of the analyzed molecules might be located in intermembrane spaces, displaying distinct orientation behavior compared with the fraction bound within the membrane [20,21]. Further, in order to precisely determine the orientation of a xanthophyll chromophore within the lipid membrane using the absorption spectroscopy method, one has to employ linear dichroism analysis. However, this necessitates the use of relatively high concentrations of the polyenes compared with lipids or the study of model systems consisting of many dozens of stacked lipid bilayers. The increased concentrations of xanthophylls above 5 mol % leads to their partial aggregation within the lipid phase [29], significantly impacting the accuracy in determining the orientation of individual molecules. Therefore, the lipid multi-bilayer systems approach is not well suited to the determination of the orientation of xanthophyll molecules within the lipid membrane. Recently, the fluorescence imaging method was used to give new insight into this problem [29]. In contrast with the studies carried out in a lipid multi-bilayer system, this technique has provided information on the localization and orientation of xanthophylls in a single lipid bilayer membrane (giant unilamellar vesicle (GUV)). By employing linearly polarized excitation light, it becomes possible to selectively excite dyes that are embedded within the lipid membrane. The absorption of light (*A*), and thus the subsequent fluorescence, depends on the angle between the electric vector of the light ($\vec{E}$) and the transition dipole moment of the Lut molecule ($\vec{M}$). This phenomenon leads to minimal fluorescence intensity in the upper and lower regions of the liposome, while the sides exhibit the highest emission values. This is also reflected in the fluorescence anisotropy parameter, which reaches its maximum value on the sides of the liposome and can be quantified on a scale from 0 to 0.4 [30,31]. The angles obtained between the transition dipole and the normal lipid bilayer formed from DMPC (dimyristoylphosphatidylcholine) are very similar for Lut and Zea, ~42° and ~43°, respectively, indicating transmembrane orientation in the membrane lipid matrix [29]. Additionally, the angle value obtained for Zea in the single DMPC membrane is significantly larger than in the lipid multi-bilayer system composed of the same lipid (~43° versus ~25°) [20]. In this context, it should be noted that the vector of the dipole moment of the Lut molecule ($\vec{M}$) is not parallel to the long axis of the polyene chain of the Lut molecule. Indeed, it has been shown that the transition dipole moment ($\vec{M}$) for linear polyenes with 10 conjugated double bonds is oriented about ~13° [32] to the long axis of the molecule chain (Figure 2). 

The confocal fluorescence microscopy used for a single unilamellar liposome reports the orientation of Lut with higher precision [29] than the angle calculated by means of linear dichroism measurements applied to the planar lipid membrane systems [21]. The transmembrane orientation of the Lut molecule in a single lipid bilayer was also confirmed in the DPPC (dipalmitoylphosphatidylcholine) lipid bilayer (Figure 3) [33]. Thus, measurements of fluorescence anisotropy are a powerful tool in the confirmation that Lut is vertically oriented to the surface of the lipid membrane. 

## 3. Lutein as Modifier of Physical Properties of the Lipid Bilayers

The low Lut content of the lipid model system is the justification to understand how the biomembrane itself affects the organization of pigments in the lipid matrix, including the organization of Lut molecules in lipid bilayer discussed previously. On the other hand, the fact that the system has a high Lut content is very important to understanding how Lut molecules affect the physical properties of the membrane. In our research, we have used various techniques to study the effect of Lut on the physical properties of the lipid bilayers. However, most of our results have been obtained using EPR spin-label methods [34,35,36], as we believe they are applicable to understanding lipid properties. The membrane properties, which can be determined by analyzing EPR spectra and saturation recovery signals, include oxygen solubility and diffusion, hydrophobicity, order parameter, fluidity, alkyl chain bending, and penetration of metal-ion complexes [24,25,26,34,35,37]. Lipid spin labels are probes that are covalently attached to lipid molecules at different positions on the alkyl chain (see examples in Figure 4). In this context, EPR spectra and saturation recovery signals recorded at different depths in the lipid membranes from the polar headgroup region to the membrane center enable the monitoring of the physical properties across membrane. Moreover, the bimolecular collision of the nitroxide fragment with useful paramagnetic probes—molecular oxygen [34,38,39] or metal–ion complexes [25,40]—additionally provides information about the spatial (three-dimensional) organization of the membrane and dynamics of lipids around the nitroxide group. 

A growing body of literature, including our own papers, has determined that the packing effect of polar carotenoids on the lipid bilayer is similar to the effect of cholesterol on membrane properties [19,35,37,41,42,43,44]. Rohmer et al. first proposed that carotenoids may play the same role of membrane stabilizers in prokaryotes as sterols play in eukaryotes [45]. The macular carotenoids terminated by two polar groups indicate similarities to the cholesterol molecule with regard to their impacts on membrane dynamics. Although, this similarity is very general, we have found that significant differences exist between these two membrane modifiers, resulting from the different molecular structures of these two modulators on membrane properties [35,37,46]. EPR studies indicate that the presence of a high amount of Lut in the lipid membrane drastically changes the membrane’s physical properties and also impacts the penetration of small, nonpolar molecules and chemical reactions occurring in the membrane [26,35,37,46]. There is evidence that the membrane’s oxygen concentration and diffusion play a critical role in chemical reactions occurring within the lipid environment. The spin-label oximetry method allows us to obtain the oxygen diffusion–concentration product (named the oxygen transport parameter) from the saturation recovery EPR measurements [34,38,39]. It should be noted here that the oxygen transport parameter is greater at all positions in the lipid bilayer than in the surrounding aqueous phase. The oxygen transport parameter for pure DMPC near position C5 of the lipid alkyl chain in the presence of cholesterol or Lut at high content in the lipid membrane is larger than for the lipid bilayer with modifier molecules. Moreover, Lut at a concentration of 5 mol% has an effect similar to cholesterol at 10 mol% (Figure 5). Generally, Lut decreases the frequency of alkyl-chain bending at all depths in the lipid bilayer and reduces the oxygen transport parameter at all membrane depths [37]. In contrast with Lut, the incorporation of cholesterol into the lipid membrane increases the packing density of the alkyl chains near the polar headgroup region, and increases the frequency of chain-bending in the membrane center [35,37], which creates hydrophobic channels for oxygen molecules in the inner core region of the lipid bilayer. Due to its mismatch with phospholipids molecules, cholesterol creates more vacant pockets in the membrane center region in which oxygen molecules may reside. This difference in the effect of Lut and cholesterol on alkyl chain bending is the result of differences in the molecular structures of these modifier molecules. A cholesterol-rigid, planar molecule with four fused rings and a side chain is located in one half of the lipid bilayer. Lut, containing an extended conjugated double-bond system, acts like a rigid rod-like molecule that spans the membrane by anchoring two hydroxyl groups on the opposite side of the lipid bilayer. 

The incorporation of the Lut into a membrane not only alters the permeability of oxygen (Figure 5) but also may result in the inhibition of the penetration of various ions toward the membrane center. Measuring the spin-lattice relaxation time in the presence and absence of fast-relaxing species such as a paramagnetic metal–ion complex allows us to obtain the penetration profiles of these molecules toward the membrane center. This situation is shown in Figure 6a for a pure DMPC membrane and a DMPC membrane with 10 mol % Lut. The penetration parameter for iron complex (${\mathrm{Fe}(\mathrm{CN})}_{6}^{-3}$) gradually decreases toward the membrane center and is lowered by incorporating Lut at all locations in the lipid bilayer. Additionally, the EPR spin-label method is useful for monitoring the local membrane’s hydrophobicity around the nitroxide moiety [25,35,48,49]. The local water penetration into the membrane can be monitored using the Z-component of the hyperfine interaction tensor of the nitroxide spin label (2*A*_Z_ value). The 2*A*_Z_ parameter is related to the dielectric constant (ε) of the medium where the spin label probe is located and has been used as a hydrophobicity parameter [35,48]. The lipid bilayer structure possesses a depth-dependent hydrophobicity gradient largely resulting from the extent of water penetration into bilayer. Hydrophobicity gradually increases toward the center of the membrane. Polar carotenoids significantly increase the hydrophobicity of the membrane core but decrease hydrophobicity in the polar region [25]. Figure 6b shows the effect of different amounts of Lut on the hydrophobicity of the DMPC membrane center measured with the 2*A*_Z_ parameter of 16-SASL (spin label with the nitroxide attached at the 16th carbon). Intercalation of 10 mol% Lut increases the hydrophobicity in the central region of the DMPC bilayer from 2*A*_Z_ = 67.63 G to 2*A*_Z_ = 65.25 G, which means the dielectric constant of the environment around the spin label changes from ε = 10 to ε = 3 [25,35]. Thus, Lut—like other dipolar carotenoids—increases hydrophobicity in the center of the membrane (Figure 6b) and consequently lowers water penetration into that region and decreases the penetration of the small polar molecule toward the membrane interior (Figure 6a). 

## 4. Lutein as a Blue Light Filter

Because the highest concentration of macular carotenoids is in the prereceptoral layers of the retina, their main role has been proposed to be the filtering of blue light [50,51,52]. This prereceptoral blue light screening by macular xanthophylls probably plays an important role in the observation of two entopic phenomena, Haidinger’s brushes and Maxwell’s spot, both of which are seen only under highly specific conditions. Maxwell’s spot is an entopic phenomenon that appears as a reddish spot in the central visual field when a white surface is viewed through a dichroic filter transmitting red and blue lights [53,54]. The perception of Maxwell’s spot may also be seen when the blue light is flickering or the blue disc is rotating. In this case, it is observed as a dark shadow at the fixation point. The entoptic Haidinger phenomenon appears if a white surface is viewed through a polarizing filter and is usually described as two perpendicularly crossed hourglass-shaped figures, one yellowish and the other blueish. The wavelength of polarized light for which the maximum intensity is shown in the perception of Haidinger’s brushes is 460 nm [55], which is also the maximum absorption for macular carotenoids (see Figure 1). One possible explanation for Haidinger’s brushes is that they are generated by the differential absorption of polarized light by radially arranged chromophores within the macula lutea. The Haidinger’s brush phenomenon, manifested by different absorptions of polarized light by the human macula, is explained by some authors as evidence for the regular arrangement of macular carotenoids in the central retina [55,56]. This explanation would not be possible without the regular packing of carotenoid molecules in radially extending fibers. In contrast with carotenes available in high amounts in the diet and plasma, xanthophylls dissolved in lipid membranes are oriented regularly across the lipid bilayer, and their orientation is vertical to the bilayer surface, which is equal for Lut and for Zea. However, it cannot be ruled out that this regular distribution is possible due to two binding proteins. Interestingly, the foveal localization of macular carotenoids and the perception of Haidinger’s brush phenomenon are correlated. Moreover, the higher macular pigment optical density leads to better visibility of Haidinger’s brushes and disturbances in macular pigment distribution observed with patients with macular telangiectasia type 2 due to a lack of Haidinger’s brush phenomena [57]. In the 1980s, Bone and Landrum were among the first to support the conclusion that macular xanthophylls exhibit dichroic properties when they are naturally oriented in the lipid bilayer in the macula lute [2,58]. The transmembrane orientation of Lut shown in Figure 3a (Zea also adopts the same vertical orientation in the membrane) in a single lipid bilayer and the radial symmetry of the axons in the Henle fiber layer leads to a highly organized structure of the blue-light-absorbing chromophores necessary for the observation of the brushes, as discussed previously.

## 5. Conclusions

This review focuses on Lut–lipid membrane interactions. Two orientations of Lut molecules (parallel and vertical to the surface of the lipid bilayer [20,21]) were predicted earlier via linear dichroism studies of the macular xanthophylls incorporated into the system of lipid multi-bilayers placed on the glass. On the contrary, Zea, the second macular pigment, was found to always adopt a vertical orientation. Unfortunately, the linear dichroism method that is used for the oriented multi-bilayer phospholipid system modified by carotenoids on solid support has a number of pitfalls. In this method, the calculated angle between the transition dipole moment ($\vec{M}$) and the axis normal to the plane of the lipid membrane does not give real information about the orientation of carotenoid molecules in the lipid bilayer, especially if some pools of pigments are localized in the area between the staked bilayers, and not exclusively inside the lipid membrane [20,21]. The confocal fluorescence microscopy used for a single lipid bilayer with carotenoid fluorophores (see Section 2) allows for determination of the real orientation of macular pigments in the model membrane [29]. The presence of high anisotropy values on the sides of the GUV (Figure 3a) may be attributed to the parallel alignment of the Lut molecules with respect to the polarization of the incident light’s electric vector ($\vec{E}$). This alignment is a consequence of the dipole moment ($\vec{M}$) of the Lut molecule lying almost on the axis of the molecule, as well as the photoselection effect demonstrated in Figure 3 (panel b). As a result of this molecular orientation, the fluorescence microscopy analysis clearly reveals a substantial increase in fluorescence intensity along the lateral sides of the GUV, while the top and bottom regions of the liposome exhibit a complete absence of fluorescence emission. The observed zero intensity values in these regions arise from the mutual perpendicularity between the dipole moment ($\vec{M}$) of the carotenoid molecule’s transition and the electric vector ($\vec{E}$) of the excitation light. A hypothetical scenario involving a horizontal arrangement of Lut on the membrane would lead to an opposite outcome. Specifically, we would expect to observe high anisotropy at the top and bottom of the GUV, along with elevated fluorescence intensity in these regions. In contrast, the sides of the GUV would exhibit low anisotropy values and nearly zero fluorescence intensity. These findings underscore the significant influence of molecular orientation on the observed anisotropy and fluorescence behavior of Lut within GUVs. The method has already been validated on other fluorescent molecules, such as amphotericin B polyene and Nile blue [30].

The fluorescence measurement shows that the angle between the dipole moment ($\vec{M}$) of the carotenoid molecule and the normal axis to the liposome membrane is the same for both macular pigments. This indicates the same transmembrane orientation for Lut and Zea. Furthermore, the EPR spin-labeling method used to study the carotenoid–membrane interaction also did not demonstrate the differences between Lut and Zea [25,35,37,46]. However, very significant differences are observed between nonpolar and macular carotenoids (dipolar carotenoids) [26]. In our previous papers, we outlined that Lut and Zea are selectively present in the retina because they have two hydroxyl functional groups attached to the rings, which contribute to their (1) high membrane solubility, (2) high chemical stability, and (3) vertical orientation in the lipid membrane [35,59]. Additionally, their impact on the physical properties of lipid membrane is strong, similar to that of cholesterol molecules, and makes important contributions to the physical state of the lipid bilayer. It should be noted here that all information about the effect of Lut on microenvironmental factors such as the hydrophobicity, oxygen concentration, diffusion product, and vertical fluctuations of lipid alkyl chains (presented in Section 3) plays a critical role in the mechanisms of chemical reactions occurring within the membrane environment. Lut and Zea decrease the oxygen diffusion–concentration product and, in the same way, also reduce the penetration of the singlet oxygen into the center of the membrane. Additionally, both macular pigments reduce fluctuations at the ends of the alkyl chains in those regions of the lipid bilayer to which the rigid portion of the carotenoid molecule extends. Finally, the penetration of paramagnetic metal ions and their complexes into the membrane interior is also inhibited in the presence of Lut and Zea in the lipid bilayer. All these effects of Lut and Zea on the physical properties of the membrane make it less susceptible to oxidation. It was recently demonstrated that in the membrane domain structure, Lut and Zea are excluded from membrane domains enriched in saturated lipids and cholesterol and are concentrated in the domain, which is enriched in unsaturated lipids susceptible to oxidation [60,61,62,63]. This xanthophyll–membrane interaction plays an important role in the protection of membrane-sensitive molecules (highly unsaturated lipids) by co-localizing them with protective xanthophylls (lipid-soluble antioxidants). However, no significant differences were found between Lut and Zea in their interactions with membranes. This similarity between Lut–membrane interactions and Zea–membrane interactions does not answer why those two pigments are heterogeneously distributed in the area of the macula (low Lut/Zea ratio in the central fovea, high Lut/Zea ratio in the peripheral part). Thus, extreme caution must be taken in future carotenoid research to explain the horizontal distribution of non-homogeneous macular carotenoids in the retina.

## Figures and Tables

**Figure 1 ijms-24-12948-f001:**
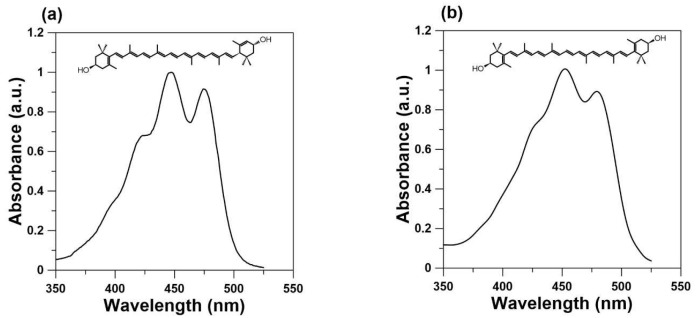
Chemical structures of (**a**) lutein and (**b**) zeaxanthin and their absorption spectra in ethanol.

**Figure 2 ijms-24-12948-f002:**
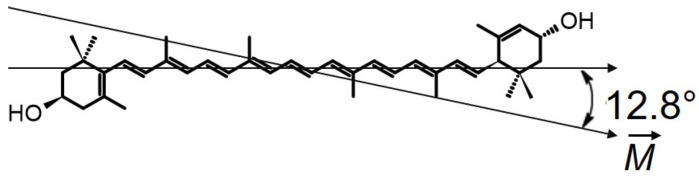
A linear 10-conjugated-double-bond system of a Lut molecule. The arrow schematically illustrates the transition dipole moment ($\vec{M}$). The calculated value of the off-axis transition dipole angle for linear conjugated polyene (N = 10) is given in Ref. [32].

**Figure 3 ijms-24-12948-f003:**
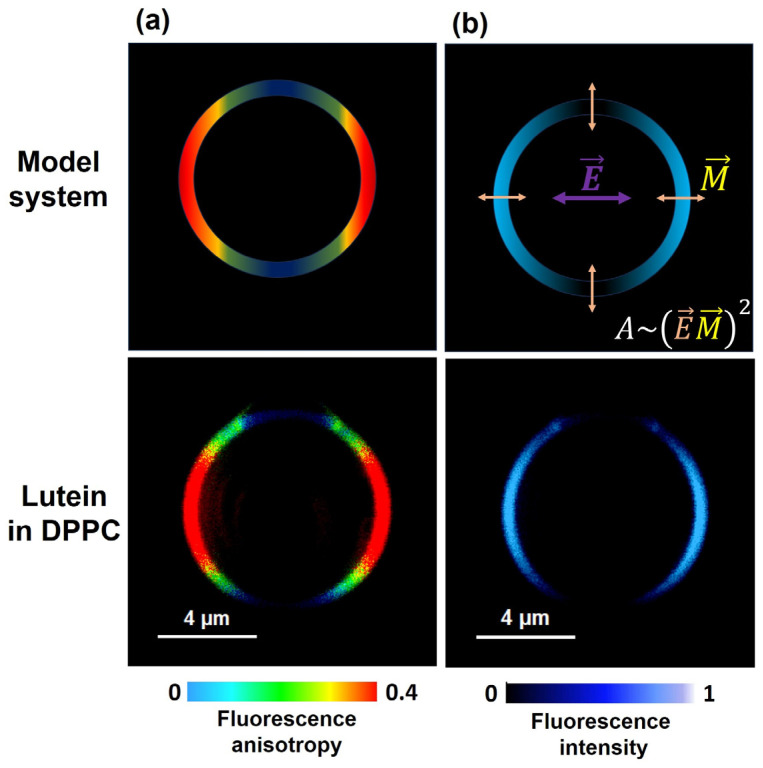
(**a**) Schematic picture of GUV with Lut. The direction of the electric vector of the probing light ($\vec{E}$) and the transition dipole moment of the Lut molecule ($\vec{M}$) are shown in the image. (**b**) The microscopic imaging of single DPPC lipid vesicles containing Lut incorporated to the lipid phase with 0.5 mol %. The left panels show fluorescence anisotropy. The right panels show fluorescence intensity. Data for Figure 3b are adapted from [33].

**Figure 4 ijms-24-12948-f004:**
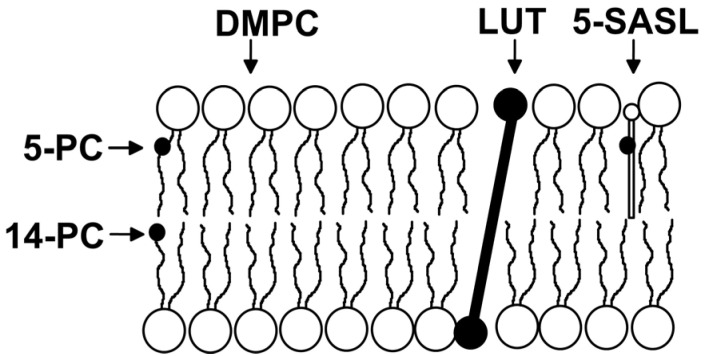
Schematic drawing showing the orientation of Lut and approximate location of the lipid spin labels (1-palmitoyl-2-(5-doxylstearoyl)phosphatidylcholine [5-PC], 1-palmitoyl-2-(14-doxylstearoyl)phosphatidylcholine [14-PC], and 5-doxylstearic acid [5-SASL]) in the DMPC bilayer. Black dots indicate the nitroxide moiety of spin labels.

**Figure 5 ijms-24-12948-f005:**
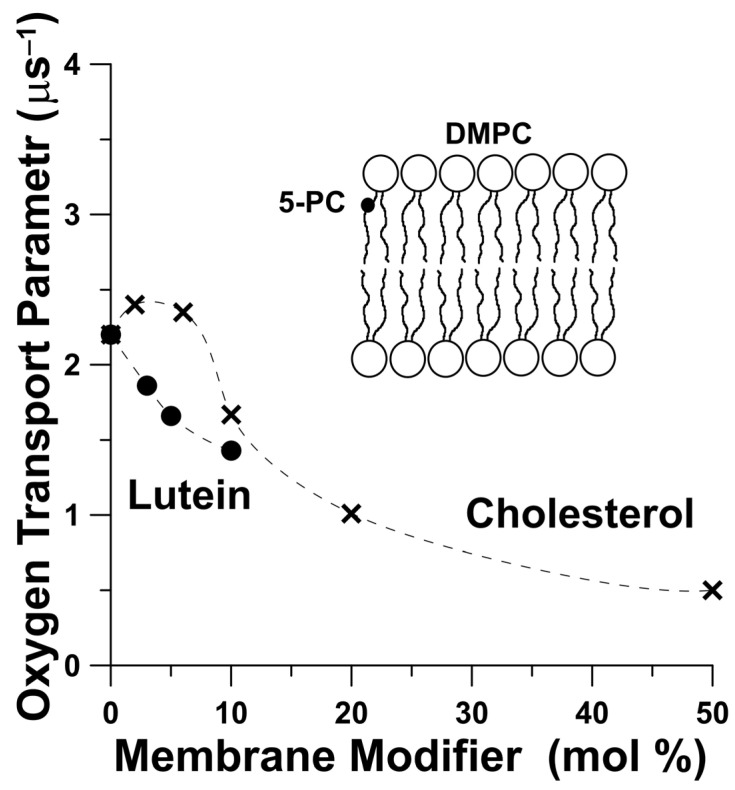
Relative oxygen transport parameter obtained with the lipid spin label 5-PC in DMPC membranes at 30 °C containing cholesterol (×) and Lut (●) plotted as a function of mol % of the modifiers of the physical properties of the lipid membranes. Data for Figure 5 are adapted from [46,47]. The schematic drawing shows the approximate location of the 5-PC lipid spin label in the DMPC bilayer. The black dot in the schematic drawing indicates the nitroxide moiety of the spin label.

**Figure 6 ijms-24-12948-f006:**
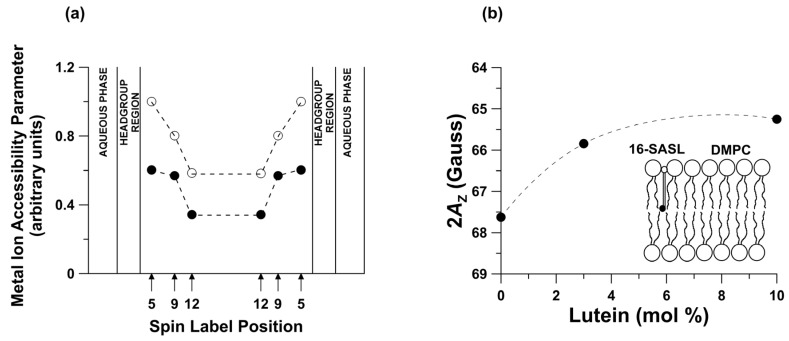
(**a**) Relative metal-ion complex accessibility parameter obtained at 30 °C in DMPC bilayer containing 0 (◯) and 10 mol% Lut (●) as a function of the position of the nitroxide moiety of 5- SASL, 9- SASL, and 12- SASL in the lipid bilayer. (**b**) Local hydrophobicity (2*A*z value) for 16- SASL in the DMPC bilayer plotted as a function of a mole fraction of Lut. Upward changes indicate decreased water penetration. The schematic drawing of the lipid bilayer shows the approximate location of the 16- SASL lipid spin label in the DMPC bilayer. The black dot in the schematic drawing indicates the nitroxide moiety of spin label. Data for Figure 6 are adapted from [25].

## Data Availability

The data presented in this review are available in [25,33,46,47].

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
