# Peer review of "An Overview of Lutein in the Lipid Membrane"

_ijms, 2023, doi:10.3390/ijms241612948_

Round 1

Reviewer 1 Report

Review report

Journal: IJMS (ISSN 1422-0067)

Manuscript ID: ijms-2505448

Type: Review

Title: An overview of lutein in the lipid membrane

Authors: Justyna Widomska *, Witold K. Subczynski , Renata Welc Stanowska , Rafał Luchowski

Section: Bioactives and Nutraceuticals

Special Issue: The Role of Lutein for Human Health

General

In this manuscript, the authors review into the interaction between lutein (Lut) and lipid membranes. They highlight the structural differences between Lut and zeaxanthin (Zea), two carotenoids selectively accumulated in the fovea of the eye. They also investigate the orientation of Lut in the lipid bilayer according to previous works. Previous studies suggested that Lut could adopt two orientations—horizontal and vertical—to the lipid membrane. However, the authors point out that recent studies using electron paramagnetic resonance (EPR) and molecular modeling suggest that Lut and Zea adopt an orientation perpendicular to the plane of the membrane.

The authors further discuss the effect of Lut on the physical properties of the lipid bilayers. They mention that Lut molecules affect the physical properties of the membrane, including oxygen solubility and diffusion, hydrophobicity, order parameter, fluidity, alkyl chain bending, and penetration of metal-ion complexes.

The authors also review the Haidinger's brushes phenomenon, which is explained by some authors as evidence for the regular arrangement of macular carotenoids in the central retina. They suggest that this regular distribution is possible due to two binding proteins.

In conclusion, in this review, the authors emphasize that Lut and Zea are selectively present in the retina because they have two hydroxyl functional groups attached to the rings, which contribute to their unique interactions with lipid membranes.

Minor points

The manuscript seems to be well-composed, adhering to a logical narrative, albeit intermittently. The authors explore into the pertinent literature regarding the interaction of Lutein with cellular membrane models. However, many of the conclusions are predicated on a limited number of studies, and there is a notable emphasis on the authors' own work, which somewhat narrows the scope of the analysis. Despite this, the manuscript provides valuable insights for readers seeking to further their understanding of the biophysical aspects of Lutein, which is highly beneficial. I would recommend its publication upon addressing the following points:

1.    On lines 23-24, the authors should revise the sentence to be more specific about the properties they are referring to, especially as this is in the "Abstract" section.

2.    I strongly suggest replacing the phrases "In the present paper…" or "In this paper…" (which appear on lines 51, 68, 322) with "In this review…". Please bear in mind that this is not an original research paper, and as such, could lead to misunderstandings.

3.    On lines 68-69, the authors state that the aim of this review is to summarize their own research, which is not the ultimate purpose of IJMS reviews. Moreover, later in the manuscript, the authors delve into and summarize the results of other authors, which is inconsistent with the statement made on lines 68-69. It is suggested that these lines be modified to better reflect the work carried out.

4.    On line 84, the authors declare: "The conclusion that Lut has two orientations horizontal and vertical to the lipid membrane is derived from a paper by Sujak et al.” This sentence should be revised, as strictly speaking, no explanation of a natural phenomenon is derived from the existence of a paper or a piece of work. Please be more precise and detail the reasoning behind this.

5.    The authors mention the term "lipid multi-bilayer” (line 121). Are they referring to "multilamellar liposomes (MLV)”? If so, please revise the sentence and terminology. The same applies to the term "single lipid bilayer membrane” (line 129), are they referring to "large unilamellar vesicles (LUV)”?

6.    On line 124 the authors state: "Therefore, the lipid multi-bilayer systems do not seem well suited for precise determination of the orientation of xanthophylls within the lipid membrane.” The authors should explain their reasons for writing this sentence. Moreover, the phrase "do not seem well" sounds rather unclear. Please provide a better justification for this statement.

7.    On line 135 the authors write: "This also is reflected in the fluorescence anisotropy parameter, which reaches its maximum value on the sides of the liposome and can be quantified on a scale from 0 to 0.4.” However, no reference is provided to support these numerical values. Please provide references.

Author Response

We appreciate the time and effort that you dedicated to providing feedback on our manuscript and are grateful for the insightful comments on and valuable improvements to our paper. The change in text appears in yellow in the revised manuscript.

C: On lines 23-24, the authors should revise the sentence to be more specific about the properties they are referring to, especially as this is in the "Abstract" section.

A: As recommended, we have changed the sentence (lines 23-24) to read: “how carotenoids affect membrane physical properties—such as fluidity, polarity, and order—in relation to membrane structure.”

C: I strongly suggest replacing the phrases "In the present paper…" or "In this paper…" (which appear on lines 51, 68, 322) with "In this review…". Please bear in mind that this is not an original research paper, and as such, could lead to misunderstandings.

A: The appropriate corrections have been made.

C: On lines 68-69, the authors state that the aim of this review is to summarize their own research, which is not the ultimate purpose of IJMS reviews. Moreover, later in the manuscript, the authors delve into and summarize the results of other authors, which is inconsistent with the statement made on lines 68-69. It is suggested that these lines be modified to better reflect the work carried out.

A: We have modified the end of Sec.1. We have changed the sentence to read:This review summarizes Lut-lipid model studies and the unique Lut-membrane interaction that distinguishes Lut and Zea from other carotenoids available in the human diet.”

C: On line 84, the authors declare: "The conclusion that Lut has two orientations horizontal and vertical to the lipid membrane is derived from a paper by Sujak et al.” This sentence should be revised, as strictly speaking, no explanation of a natural phenomenon is derived from the existence of a paper or a piece of work. Please be more precise and detail the reasoning behind this.

A: This sentence was changed in the revised manuscript. Now the sentence reads as follows: “Two orientations of the Lut molecule—horizontal and vertical to the lipid bilayer—were previously reported by Sujak et al.”

C: The authors mention the term "lipid multi-bilayer” (line 121). Are they referring to "multilamellar liposomes (MLV)”? If so, please revise the sentence and terminology. The same applies to the term "single lipid bilayer membrane” (line 129), are they referring to "large unilamellar vesicles (LUV)”?

A: The term "lipid multi-bilayer” does not refer to MLV but to the supported lipid bilayer system (SLB) whereas "single lipid bilayer membrane” refers to giant unilamellar vesicle (GUV). We have revised this passage to read: “In the model system comprising supported lipid bilayers, the orientation angle depends on the number of stacked bilayers.” … “In contrast to the studies carried out in a lipid multi-bilayer system, this technique has provided information on the localization and orientation of xanthophylls in a single lipid bilayer membrane (giant unilamellar vesicle [GUV]).”

C: On line 124 the authors state: "Therefore, the lipid multi-bilayer systems do not seem well suited for precise determination of the orientation of xanthophylls within the lipid membrane.” The authors should explain their reasons for writing this sentence. Moreover, the phrase "do not seem well" sounds rather unclear. Please provide a better justification for this statement.

A: As suggested, we added an explanation why the lipid multi-bilayer systems do not always give the correct  the angle of orientation of the carotenoid molecules in a stacked lipid membrane system. In the revised manuscript, we have added this explanation. We have changed the text in lines 124-126. Now, these sentences read as follows: “In the model system comprising supported lipid bilayers, the orientation angle depends on the number of stacked bilayers. This is most likely due to the fact that a certain fraction of the analyzed molecules might be located in intermembrane spaces, displaying distinct orientation behavior compared with the fraction bound within the membrane [20,21]. Further, in order to precisely determine the orientation of a xanthophyll chromophore within the lipid membrane by the absorption spectroscopy method, one must employ a linear dichroism analysis. However, this necessitates the use of relatively high concentrations of the polyenes compared with lipids or the study of model systems consisting of many dozens of stacked lipid bilayers. The increased concentrations of xanthophylls above 5 mol % leads to their partial aggregation within the lipid phase [29], significantly impacting the accuracy in determining the orientation of individual molecules. Therefore, the lipid multi-bilayer systems approach is not well suited for determination of the orientation of xanthophyll molecules within the lipid membrane. Recently, the fluorescence imaging method was used to give new insight into this problem [29]. In contrast with the studies carried out in a lipid multi-bilayer system, this technique has provided information on the localization and orientation of xanthophylls in a single lipid bilayer membrane (giant unilamellar vesicle [GUV]).”

  1. Gruszecki, W.I.; Sielewiesiuk, J. Orientation of Xanthophylls in Phosphatidylcholine Multibilayers. Biochim. Biophys. Acta 1990, 1023, 405–412, doi:10.1016/0005-2736(90)90133-9.
  2. Sujak, A.; Gabrielska, J.; GrudziÅ„ski, W.; Borc, R.; Mazurek, P.; Gruszecki, W.I. Lutein and Zeaxanthin as Protectors of Lipid Membranes against Oxidative Damage: The Structural Aspects. Arch. Biochem. Biophys. 1999, 371, 301–307, doi:10.1006/abbi.1999.1437.
  3. Grudzinski, W.; Nierzwicki, L.; Welc, R.; Reszczynska, E.; Luchowski, R.; Czub, J.; Gruszecki, W.I. Localization and Orientation of Xanthophylls in a Lipid Bilayer. Sci. Rep. 2017, 7, 9619, doi:10.1038/s41598-017-10183-7.

C: On line 135 the authors write: "This also is reflected in the fluorescence anisotropy parameter, which reaches its maximum value on the sides of the liposome and can be quantified on a scale from 0 to 0.4.” However, no reference is provided to support these numerical values. Please provide references.

A: The appropriate references are now included.

  1. Grudzinski, W.; Sagan, J.; Welc, R.; Luchowski, R.; Gruszecki, W.I. Molecular Organization, Localization and Orientation of Antifungal Antibiotic Amphotericin B in a Single Lipid Bilayer. Sci. Rep. 2016, 6, 32780, doi:10.1038/srep32780.
  2. Fluorescence Anisotropy. In Principles of Fluorescence Spectroscopy; Lakowicz, J.R., Ed.; Springer US: Boston, MA, 2006; pp. 353–382 ISBN 978-0-387-46312-4.

Reviewer 2 Report

It is an interesting review.

I have several comments:

1)Figures 3,4,5,6 do not have proper description of symbols (dots, x, and others).

2)"The confocal fluorescence microscopy used for a single lipid bilayer with carotenoid fluorophore (see Sec. 3) allows for determination of the real orientation of macular pigments in the model membrane."

Please describe why is it real? What is the model? How do you compare model and sample? How it would look if lutein would be located horizontally? What about zeaxanthin? Give othe examples of using this method with proven results. It looks it is all based on one publication and it is not enough to say it is a real orientation. Please corroborate.

Author Response

We appreciate the time and effort that you dedicated to providing feedback on our manuscript and are grateful for the insightful comments on and valuable improvements to our paper. The change in text appears in yellow in the revised manuscript

C: Figures 3,4,5,6 do not have proper description of symbols (dots, x, and others).

A: All these figures were corrected.

C: The confocal fluorescence microscopy used for a single lipid bilayer with carotenoid fluorophore (see Sec. 3) allows for determination of the real orientation of macular pigments in the model membrane." Please describe why is it real? What is the model? How do you compare model and sample? How it would look if lutein would be located horizontally? What about zeaxanthin? Give othe examples of using this method with proven results. It looks it is all based on one publication and it is not enough to say it is a real orientation. Please corroborate.

 A: Thank you for your valuable questions that greatly improve the clarity of our work. Following is an explanation in response to the questions raised; it has been added also to the text of the manuscript: “The confocal fluorescence microscopy used for a single lipid bilayer with carotenoid fluorophore (see Sec. 2) allows for determination of the real orientation of macular pigments in the model membrane [29]. The presence of high anisotropy values on the sides of the GUV (Fig. 3a) may be attributed to the parallel alignment of the Lut molecules with respect to the polarization of the incident light's electric vector (E). This alignment is a consequence of the dipole moment (M) of the Lut molecule lying almost on the axis of the molecule, as well as the photoselection effect demonstrated in Fig. 3 (panel b). As a result of this molecular orientation, the fluorescence microscopy analysis clearly reveals a substantial increase in fluorescence intensity along the lateral sides of the GUV, while the top and bottom regions of the liposome exhibit a complete absence of fluorescence emission. The observed zero intensity values in these regions arise from the mutual perpendicularity between the dipole moment (M) of the carotenoid molecule's transition and the electric vector (E) of the excitation light. A hypothetical scenario involving a horizontal arrangement of Lut on the membrane would lead to an opposite outcome. Specifically, we would expect to observe high anisotropy at the top and bottom of the GUV, along with elevated fluorescence intensity in these regions. In contrast, the sides of the GUV would exhibit low anisotropy values and nearly zero fluorescence intensity. These findings underscore the significant influence of molecular orientation on the observed anisotropy and fluorescence behavior of Lut within GUVs. The method has already been validated on other fluorescent molecules, such as amphotericin B polyene and nile blue [30].”

  1. Grudzinski, W., J. Sagan, R. Welc, R. Luchowski and W. I. Gruszecki. "Molecular organization, localization and orientation of antifungal antibiotic amphotericin b in a single lipid bilayer." Sci Rep 6 (2016): 32780. 10.1038/srep32780.

Reviewer 3 Report

The paper is well prepared and written. The authors extensively reviewed the structure and function of Lut and Zea. Although the major contents are not new, yet the scope the author covered is quite extensive. One major issue for the paper is that the authors should cite more new papers in the field.

Minor modifications might be needed 

Author Response

C: The paper is well prepared and written. The authors extensively reviewed the structure and function of Lut and Zea. Although the major contents are not new, yet the scope the author covered is quite extensive. One major issue for the paper is that the authors should cite more new papers in the field.

A: Thank you for this suggestion. It would have been interesting to explore new papers in this scientific area. However, to our knowledge, most of the references related to the influence of Lut on model membrane physical properties have been cited. A goal of this review is to provide detailed information about the effect of Lut on the lipid membrane. EPR spin-labeling methods provide a number of unique approaches for determining several important membrane properties, such as hydrophobicity, oxygen solubility and diffusion rates, membrane order, and fluidity. Subczynski’s laboratory focuses on these areas (Refs. 16, 24, 25, 26, 32–38, 44–46, 57). Additionally, confocal fluorescence microscopy studies of giant unilamellar vesicles (simple model membrane systems) allow the angle of the transition dipole moment of the Lut molecule located in the lipid bilayer to be obtained, and give information about the orientation and organization of Lut in the lipid matrix (Refs. 29, 31). Gruszecki’s laboratory is focused on the biophysical aspects of carotenoids in biomembranes and uses florescence and Raman imaging methods. Finally, molecular dynamics simulation also sheds light on the structural properties of lipid bilayers (Refs. 22, 27, 28). As you suggested, we added new references to the Conclusion section. The added sentences reads: “It was recently demonstrated that in the membrane domain structure, Lut and Zea are excluded from membrane domains enriched in saturated lipids and cholesterol and are concentrated in the domain, which is enriched in unsaturated lipids susceptible to oxidation [60-63]. This xanthophyll-membrane interaction plays an important role in the protection of membrane-sensitive molecules (highly unsaturated lipids) by co-localizing them with protective xanthophylls (lipid-soluble antioxidants).” We believe that all new papers in the Lut-lipid membrane interaction field are now cited in our review. The change in text appears in yellow in the revised manuscript. In addition to your comment, the paper was corrected by a native English speaker, a Scientific Copy Editor at the Department of Biophysics, Medical College of Wisconsin. These corrections are not highlighted.

  1. Wisniewska-Becker, A., G. Nawrocki, M. Duda and W. K. Subczynski. "Structural aspects of the antioxidant activity of lutein in a model of photoreceptor membranes." Acta Biochim Pol 59 (2012): 119-24.
  2. Wisniewska, A. and W. K. Subczynski. "Distribution of macular xanthophylls between domains in a model of photoreceptor outer segment membranes." Free Radic Biol Med 41 (2006): 1257-65. 10.1016/j.freeradbiomed.2006.07.003.
  3. Wisniewska, A. and W. K. Subczynski. "Accumulation of macular xanthophylls in unsaturated membrane domains." Free Radic Biol Med 40 (2006): 1820-6. 10.1016/j.freeradbiomed.2006.01.016.
  4. Subczynski, W. K., A. Wisniewska and J. Widomska. "Location of macular xanthophylls in the most vulnerable regions of photoreceptor outer-segment membranes." Arch Biochem Biophys 504 (2010): 61-6. 10.1016/j.abb.2010.05.015.
